# Antimicrobial and Antifungal Action of Biogenic Silver Nanoparticles in Combination with Antibiotics and Fungicides Against Opportunistic Bacteria and Yeast

**DOI:** 10.3390/ijms252312494

**Published:** 2024-11-21

**Authors:** Artem Rozhin, Svetlana Batasheva, Liliya Iskuzhina, Marina Gomzikova, Marina Kryuchkova

**Affiliations:** 1Institute of Fundamental Medicine and Biology, Kazan Federal University, Kreml Str. 18, 420008 Kazan, Russia; svbatasheva@gmail.com (S.B.); iskuzhina.l@yandex.ru (L.I.);; 2Institute for Regenerative Medicine, Sechenov University, Trubetskaya Str. 8/2, 119992 Moscow, Russia

**Keywords:** biogenic silver nanoparticles, antibiotic, fungicide, antimicrobial action, antifungal action, *Pseudomonas putida*, *Alcanivorax borkumensis*, *Candida*

## Abstract

The development of multidrug resistance by pathogenic bacteria and yeast is a significant medical problem that needs to be addressed. One possible answer could be the combined use of antibiotics and silver nanoparticles, which have different mechanisms of antimicrobial action. In the same way, these nanoparticles can be combined with antifungal agents. Biogenic silver nanoparticles synthesized using environmentally friendly biosynthesis technology using extracts of biologically active plants are an effective nanomaterial that needs to be comprehensively investigated for implementation into medical practice. In this study, the synergistic effects arising from their combined use with antibiotics and fungicides against various bacteria and yeasts were studied. The following methods were used: disco-diffusion analysis and construction of plankton culture growth curves. The synergistic effect of silver nanoparticles and antibiotics (fungicides) has been determined. Effective concentrations of substances were established, recommendations for the studied pathogenic species were presented, and the effect of destruction of the bacterial membrane was illustrated. The most significant synergistic effect was manifested in pathogenic candida and brewer’s yeast.

## 1. Introduction

Despite the fact that oncological and cardiovascular diseases have overtaken infectious diseases in terms of mortality, the overcoming of pathogenic bacteria is still an extremely important task. Microorganisms can cause nosocomial infections in a wide variety of medical departments (surgical, intensive care, burn departments), aggravating the condition of patients. Difficulties in the treatment of these infections are caused, among other things, by the development of bacterial resistance to commonly used drugs [1]. To solve this problem, the current clinical situation is being investigated, new, improved approaches are being introduced in the use of existing drugs, and new techniques and solutions are being sought.

The main microorganisms that most often cause infections in the hospital environment are *Escherichia coli*, *Klebsiella pneumoniae*, *Enterococcus* spp., *Staphylococcus aureus*, *Staphylococcus epidermidis*, *Mycobacterium tuberculosis*, *Pseudomonas aeruginosa*, *Acinetobacter baumannii*, *Salmonella* spp., and *Burkholderia* spp. [1]. Besides these species, *Serratia marcescens* and other *Staphylococcus* species are also mentioned in the Chinese hospital [2]. In Germany, primarily *S. aureus*, *E. coli*, *P. aeruginosa,* and *K. pneumoniae* caused SARS (Severe acute respiratory syndrome). These four pathogens covered more than 50% of cases [3]. In Spain, *K. pneumoniae* were the most common, followed by *S. aureus* and *E. coli*. [4] In the burn center in Romania, the most common blood pathogens was *P. aeruginosa*, *S. aureus*, and *Klebsiella* spp. [5]. In all these cases, it is possible to note the bacteria that are included in the so-called ESKAPE list (*Enterococcus faecium*, *Staphylococcus aureus*, *Klebsiella pneumoniae*, *Acinetobacter baumannii*, *Pseudomonas aeruginosa*, and *Enterobacter* spp, as well as *Escherichia coli*). Many authors have noted the multidrug resistance of these bacteria. It was noted that the total number of multidrug-resistant bacteria from various hospital departments in China increased by more than 3 times in 2023 [2]. *S. aureus* showed resistance to azithromycin, erythromycin, clarithromycin, and penicillin, *E. coli* showed the highest percentage of ampicillin resistance (80%), *P. aeruginosa* was resistant to ciprofloxacin, gentamicin, ticarcillin, and levofloxacin, and for *A. baumannii* high resistance, too many drugs was observed [2]. In Spain, the proportion of multidrug resistance in *A. baumannii* and *P. aeruginosa* together was 15.4% [4]. In Romania, 31% of *S. aureus* strains resistant to oxacillin and 93% of isolated resistant strains of *Pseudomonas* spp. were identified; in *E. coli*, the proportion of such strains was lower—6% [5]. In Italy, over the past four years, there has been a slight increase in resistance to *A. baumannii*, *P. aeruginosa*, *K. pneumoniae*, *Enterobacter* spp., *S. aureus*, and *E. faecium* [6]. Anyway, different authors have reported the resistance of bacteria to almost all classes of antibiotics, including aminoglycosides, cephalosporins, fluoroquinolones, β-lactams, and colistin, an antimicrobial drug [1]. For example, enterobacteria, especially E. coli and K. pneumoniae, can become dangerous pathogens by becoming resistant, including through the production of carbapenemases [6]. About 90% of infections caused by pathogens with acquired resistance occurred because of empirical antibiotic treatment, with prolonged hospital stay [1]. Perhaps the increase in the incidence of antimicrobial-resistant pathogens can be explained by their overuse during the 2019 coronavirus pandemic [6]. Microorganisms are highly adaptable and rapidly evolve under the influence of new antimicrobial treatments. Thus, even new antibiotics are not enough to combat them [1]. It is necessary to search for new methods and solutions to combat resistant pathogens.

Fungal infections are also a modern problem that is far from being resolved soon. The development of resistance of pathogenic fungi to the antifungal drugs used is of concern. Yeast of the genus *Candida* is the causative agent of candidiasis, a widespread disease. In the hospital environment, *Candida albicans* and *Candida tropicalis* species are most often noted [2]. *C. albicans* has become a serious health problem, especially in immunocompromised patients, due to the antifungal drug resistance that many strains have developed. These yeasts cause 78% of fungal nosocomial infections and more than 10% of all nosocomial infections [7]. The other non-albicans *Candida* species are also of growing concern in hospital epidemiology [8]. For example, a wide range of antifungal drugs best knows *Candida auris*, first described in 2009, for its strong resistance. *C. auris* isolates are resistant to fluconazole, amphotericin B and echinocandins. About 90% of *C. auris* strains have acquired resistance to at least one antifungal drug [9]. This is manifested in the absence of in vitro growth inhibition when the minimum inhibitory concentrations of the selected drug cease to effectively suppress colony growth. This may also be evidenced by the clinical deterioration of the patient’s condition and the constant isolation of the pathogen from his clinical samples despite the administration of an antifungal drug [8]. The two main factors in the development of clinical resistance are the ability of *Candida* spp. to form a biofilm and reduce the concentration of the drug in the infected area, which allows some parts of the yeast to survive and acquire resistance [8]. In addition, the widespread use of antifungal drugs in the clinic and in agricultural conditions can serve as a possible selective force in creating a reservoir of *Candida* spp. strains resistant to antifungal drugs [9]. Brewer’s yeast (*Saccharomyces cerevisiae*) is widely used in the food industry and is considered harmless, which allows it to be used as a convenient model object in research, which can partially replace pathogenic species of the genus *Candida*. Compared to other groups of microorganisms, *S. cerevisiae* are not considered aggressive pathogens, but they are nevertheless capable of causing human diseases under certain circumstances. Several studies have analyzed the potential virulence of this type of yeast, and it has been suggested that some strains have the potential to cause disease [10]. For example, at the Center for Kidney Diseases and Transplantation at Baghdad Medical Center, 19 patients with *S. cerevisiae*-caused fungemia were found [11].

Thus, it can be summarized that there is a problem of developing resistance by pathogenic bacteria and yeast, which leads researchers to the need to find new methods and techniques that are effective in conditions when antibiotics and antifungal agents no longer work. Silver nanoparticles (AgNPs) could be such an instrument.

Silver nanoparticles are currently widely used in a variety of fields. They are used for wastewater treatment [12,13,14], in agriculture [15,16,17], as a catalytic agent [18,19,20], and in many new studies, which indicates great prospects for this nanomaterial. Most often, these nanoparticles are used in biomedicine, the reason for which is the long-known properties of silver that suppress bacterial growth [21]. They are used directly and in complexes with other substances, such as antibacterial agents [22,23,24,25,26,27], fungicides [28,29,30,31] and antiviral agents [32,33]. Considering that silver can be used to cover surfaces and make biofouling-resistant materials, nanoparticles are also used in dentistry [34,35,36], for the manufacture of wound dressings [37,38,39], and in the manufacture of prostheses and catheters [40,41,42]. There are also examples of their use against cancer due to their ability to penetrate the cell under certain conditions and induce apoptosis [43,44,45]. These data allow us to consider silver nanoparticles as a promising material that can be investigated as a possible alternative to antibiotics. In this case, the size, shape, and other characteristics of the particles, the methodological conditions of their use, and the method of their synthesis are very important.

Silver nanoparticles are currently synthesized in numerous ways: chemical (chemical reduction, microemulsion, and photochemical method), physical (laser ablation, irradiation, evaporation, condensation), and biological (using plants, bacteria, or fungi) [46]. Biological methods are considered to be the most environmentally friendly. In this case, biogenic nanoparticles are even labeled “Bio-AgNPs”. Most often, a plant extract is used for synthesis; the process takes place at a low (20–60 °C) temperature and without the use of any harmful chemicals [47,48,49,50,51]. The precursor is silver nitrate, which is reduced by a plant extract, while the nanoparticles “build up” in solution according to the “bottom-up” principle. A film of reducing agent can form on the surface of the particles, which can give them additional properties, for example, an obstacle to the formation of conglomerates [52]. Given that many researchers expect to create an antimicrobial product, they try to use plant extracts in such a synthesis, which themselves have antibacterial properties. In our study, *Sphagnum fallax* moss was chosen as such a plant. We used this type of moss as the basis for the preparation of an extract, which was used as a reducing agent in the biogenic synthesis of silver nanoparticles. *Sphagnum* mosses are known for their wound healing and bacteriostatic properties [53,54]; they are even traditionally used as a dressing material [55]. Recent studies have shown that secondary metabolites of the *S. fallax* have a pronounced antimicrobial effect [56]. The mechanism of action of this moss on bacteria has not been fully studied, but it seems to be related to a substance contained in it called Sphagnol [57].

The mechanisms of action of silver nanoparticles on a living cell differ from the action of *sphagnum* extracts and can be different: these are interaction with the cell wall, penetration through the membrane, production of reactive oxygen species, DNA damage, and inhibition of protein synthesis [58,59]. However, the effect of such an effect on the cell is not always negative. As for mammalian cells, the production of reactive oxygen species because of exposure to small silver nanoparticles can enhance the process of adipogenesis of white cells [60]. Many researchers claim that silver nanoparticles are not toxic to human cells, for example, with respect to peripheral blood mononuclear [61] or fibroblasts [62]. These data give hope that nanoparticles that are not dangerous to humans will be able to selectively destroy cells of pathogenic bacteria and yeast, for which they are toxic. Their toxicity depends on their shape, size, and surface modification [63].

Despite the abundance of promising studies on the use of silver nanoparticles to combat pathogens, their clinical use is very limited. It seems that specific data on minimum inhibitory concentrations and methodological recommendations could change this situation for the better.

In our investigation, we synthesized biogenic silver nanoparticles and studied their antimicrobial and antifungal properties, combined with effective antibiotics and fungicides. Various methods were used: disco diffusion analysis, a replicator stamp, and the construction of plankton culture growth curves. Based on these data, the minimum inhibitory concentrations were established, and the synergistic effect was studied.

## 2. Results

### 2.1. Antimicrobial and Antifungal Activity of Silver Nanoparticles

#### 2.1.1. Disco Diffusion Analysis of Various Concentrations of Silver Nanoparticles

The results of the disco diffusion analysis are shown in Table 1. The absence of a visible inhibition zone indicates the absence of the effect of the studied concentration of silver nanoparticles. The concentrate of *sphagnum* extract used for biogenic synthesis showed no results.

#### 2.1.2. Anti-Biofilm Bactericidal Effect of Silver Nanoparticles, Shown Using a Replicator Stamp

As a result, it was found that *sphagnum* extract and control drops (water) did not interfere with the growth of biofilms on agar in any variant. Figure 1 shows the most obvious results for two bacteria. The islands of pure agar on the left edge of the Petri dishes mean that the bacteria in this area were destroyed by a drop of nanoparticle suspension. *S. aureus* was resistant to all concentrations except 800 and 1000 μg/mL. *S. marcescens* and *P. putida* were resistant to silver nanoparticles, with the exception of concentrations of 1000, 800, and 600 μg/mL. *E. coli* and *A. borkumensis*, as the two species most sensitive to nanoparticles, detected vague, non-visual inhibition zones at concentrations from 50 to 1000 μg/mL. *E. coli* turned out to be more sensitive. The results correlate with the data obtained on the growth curves of these species. At the same time, there is a tendency—the bactericidal concentration of nanomaterials in this application significantly exceeds the concentration of complete inhibition of growth curves.

#### 2.1.3. Growth Curves of Planktonic Crops in the Presence of Silver Nanoparticles

The results of the analysis of the antimicrobial and antifungal activity of silver nanoparticles against the planktonic forms of *E. coli*, *S. marcescens*, *S. aureus,* and *C. albicans* are shown in Figure 2 and Figure 3. The X-axis shows the hours of the experiment, and the Y-axis shows the optical density of the suspension.

##### *Staphylococcus* *aureus*

As a result of the disco diffusion analysis, it was found that silver nanoparticles are effective against *S. aureus* at concentrations of 75 μg/mL and above (Table 1); the effect was enhanced with increasing concentration. The concentration of 50 μg/mL had no visible effect. At the same time, according to the growth curves (Figure 2), a concentration of 50 μg/mL is nevertheless capable of inhibiting colony growth. The graph shows that the growth curves in the presence of concentrations 6, 12, and 25 practically do not differ from the control curve, which indicates the absence of an antimicrobial effect. At a concentration of 50 μg/mL, the onset of colony population growth slows down by about 11 h, after which growth begins in accordance with the usual dynamics. Concentrations of 75 and 100 μg/mL significantly slow down the onset of colony population growth by 32 and 42 h, respectively, and the growth dynamics in this case are also significantly reduced. This allows us to judge the appearance of the suppressive effect of silver nanoparticles at a concentration of 50 μg/mL, as well as a significant inhibitory effect on staphylococcus, starting from concentrations of 100 μg/mL. *Sphagnum* extract had no antimicrobial effect.

##### *Escherichia* *coli*

The effectiveness of various concentrations of AgNPs against the *E. coli* bacterium has been established in experiments, the results of which are shown in Table 1 and Figure 2. Based on the disco diffusion analysis, it can be concluded that the first effective concentration is also a concentration of 50 μg/mL. With increasing concentration, the effect increases and becomes maximum at 200 μg/mL, after which it does not change. The same general patterns can be noted when analyzing the growth curves of *S. aureus*. At a concentration of 50 μg/mL, there was also a slowdown in the onset of colony population growth and growth with normal dynamics. Growth retardation was 4 h longer, up to 15 h; the same effect, with a delay of 26 h, was observed at a concentration of 75 μg/mL, and the dynamics after the delay was also normal. These concentrations temporarily inhibited the growth of *E. coli*, while at concentrations of 100 and 200 μg/mL, bacterial reproduction did not occur at all. Despite the similarity of the results with the data on *S. aureus* (namely, the minimum effective concentration is 50 μg/mL, effective inhibition is 100 μg/mL and higher), individual characteristics of the two studied species were manifested here. *E. coli* was found to be more sensitive to nanoparticles compared with staphylococcus. Colony growth at 50 μg/mL began a little later, while at 100 μg/mL, complete inhibition was observed. At the same time, it can be noted that the concentration of 75 μg/mL had a greater inhibitory effect on *S. aureus* than on *E. coli*, which indicates that the growth of the suppressive effect with an increase in the concentration of active nanoparticles does not occur gradually, and this dependence differs between different types of bacteria. *Sphagnum* extract had no antimicrobial effect.

##### *Serratia* *marcescens*

Table 1 and Figure 3 show the results of similar experiments with the bacterium *S. marcescens*. The insignificant effect of inhibition began already at a concentration of 25 μg/mL. At the same time, the concentration of 200 μg/mL was insufficient for effective inhibition in both experiments. The studied concentrations from 25 to 200 μg/mL significantly slowed down the onset of colony growth less than in the case of *S. aureus* than in *E. coli*. However, the growth dynamics (steepness of the curve in the lag phase) and the final amplitude (maximum number when reaching the plateau) gradually changed compared with the control. This may indicate that silver nanoparticles, even in small concentrations, constantly inhibit the normal growth of the *S. marcescens* colony, whereas a slowdown in the onset of colony growth followed by normal dynamics, as in *S. aureus* and *E. coli*, may indicate the proliferation of single bacteria capable of developing resistance to low concentrations of nanoparticles. At the same time, the high concentration of nanoparticles inhibited the growth of even the most resistant bacteria from the entire population. In the case of *S. marcescens*, such a concentration is probably a concentration in the range of 200 to 500 μg/mL. *Sphagnum* extract had no antimicrobial effect.

##### *Candida* *albicans*

Studying the results of similar experiments with *C. albicans* yeast, shown in Table 1 and Figure 3, it can be noted that a high concentration of nanoparticles must be used in a Petri dish to suppress the growth of plankton culture and fungal culture. The inhibition effect begins at a concentration of 100 μg/mL, while complete suppression occurs at a concentration of 300 μg/mL. The growth dynamics change significantly, and the total number decreases when reaching the plateau, while the suppression effect, unlike bacteria, manifests itself almost immediately. As in previous cases, a higher concentration of nanoparticles was required for inhibition on a solid medium than in a liquid one. This is due to the fact that when shaken, the nanoparticles enter the liquid column and interact more effectively with microorganisms than in a stationary form. *Sphagnum* extract had no significant effect on the growth of yeast culture.

### 2.2. Research on the Effectiveness of Antibiotics and Fungicides

#### 2.2.1. Comparative Disco-Diffusion Analysis of Six Antibiotics

As a result of disco-diffusion analysis, the size of the antibiotic inhibition zone (streptomycin, tetracycline, ceftriaxone, ciprofloxacin, erythromycin, and amoxicillin) was determined for *S. aureus*, *S. marcescens*, *P. putida*, *A. borkumensis,* and *E. coli*. (Figure 4). Each 6 mm disc contained 10 μL of antibiotic, with a concentration of 10 μg/mL. In subsequent studies, one most effective antibiotic was selected for each bacterium. In the further studies, we selected amoxicillin for *S. aureus* [7], ciprofloxacin for *S. marcescens*, *E. coli,* and *P. putida* [5], and ceftriaxone [3] for *A. borkumensis*.

It can be noted that the inhibition zones around antibiotic discs are much larger than around nanoparticle discs. This is due to the fact that nanoparticles are insoluble, rather heavy substances that cannot move freely over the surface of the agar under the influence of diffusion. Silver nanoparticles seem to be much more effective if they are used in the thickness of the suspension. At the same time, experiments with a replicator stamp have shown the effectiveness of their bactericidal action, even on solid agarized media, but at the same time, it is necessary to repeatedly increase the dosage.

#### 2.2.2. Comparative Disco-Diffusion Analysis of Three Fungicides

As a result of a similar disco-diffusion analysis, it was found that there was no inhibitory effect from copper sulfate and chloramine B, but Desgran inhibited all three types of yeast (Figure 5). Thus, Desgrane (tetramethylenediethylenetetramine) was chosen as an effective fungicide for all three species to build growth curves. *C. lipolytica* turned out to be the most susceptible species, *S. cerevisiae* was also sensitive to a concentration of 3.5%, and *C. albicans* inhibited only the maximum concentration—35%.

### 2.3. Growth Curves of Planktonic Crops in the Presence of Silver Nanoparticles, Together with an Antibiotic or Fungicide

#### 2.3.1. *Pseudomonas putida*

Silver nanoparticles and ciprofloxacin had different effects on the growth of the planktonic form of *P. putida*. The antibiotic reduced the amplitude and rate of colony growth without affecting the start time of colony growth. Silver nanoparticles did not decrease the amplitude and steepness in the exponential growth stage, only delaying the beginning of colony development. Thus, AgNPs completely inhibited growth at the beginning, and with the action of an antibiotic, colony growth began immediately but slowly. Their joint application was effective because both effects were superimposed on each other, and as a result, the inhibition was complete. In Figure 6, it can be seen that at extremely low concentrations of nanoparticles and an antibiotic (2.5 μg/mL and 1 μg/mL), the effect is enhanced, and growth inhibition becomes noticeable. Based on data on a number of concentrations of AgNPs (2.5 μg/mL, 5 μg/mL, 10 μg/mL and 25 μg/mL) and ciprofloxacin (0.1 μg/mL, 0.2 μg/mL, 0.3 μg/mL, 0.5 μg/mL and 1 μg/mL), the FIC I index was calculated, which it corresponded to the partial synergy of silver nanoparticles and ciprofloxacin.
FIC I = 2.5/10 + 0.1/0.2 = 0.75.

#### 2.3.2. *Alcanivorax borkumensis*

The antibiotic had a significantly lower (several times) effect on the bacterium *A. borkumensis* in comparison with the previously described *P. putida*. At the same time, the bacterium *A. borkumensis* was significantly more sensitive to silver nanoparticles than to an antibiotic. In Figure 6, it can be seen that at low concentrations, a partial addition of effects is noticeable. Different concentrations of AgNPs (0.25 μg/mL, 0.5 μg/mL, 1 μg/mL, 2.5 μg/mL and 5 μg/mL) and ceftriaxone (0.1 μg/mL, 0.5 μg/mL, 1 μg/mL, 2 μg/mL and 5 μg/mL) were studied. There was practically no synergistic effect while starting with a negligibly low concentration of silver nanoparticles (5 μg/mL); complete inhibition took place.
FIC I = 2.5/2.5 + 2/5 = 1.4 (no synergistic effect).

In this case, nanoparticles are more effective to use without the addition of an antibiotic, and they completely inhibit growth at low concentrations, which is probably due to the sensitivity of the bacterium *A. borkumensis* to them. Such an interesting effect suggests that for each new bacterium, it is necessary to select the most effective method.

#### 2.3.3. *Escherichia coli*

The antibiotic ciprofloxacin and silver nanoparticles individually have a significant inhibitory effect on *E. coli*. This is the most vulnerable species of all five studied. It is susceptible to nanoparticles like *A. borkumensis* and to an antibiotic like *P. putida*. Different concentrations of AgNPs (0.5 μg/mL, 1 μg/mL, 2 μg/mL and 5 μg/mL) and ciprofloxacin (0.01 μg/mL, 0.025 μg/mL, 0.05 μg/mL, 0.1 μg/mL and 0.5 μg/mL) were studied. There is practically no synergistic effect between the effects of these substances. The exception is the concentrations of 0.1 ciprofloxacin and 2.0 nanoparticles (Figure 7), at which the addition of effects can be noted. In other cases, there is no synergy, which is confirmed by the calculations of the index.
FIC I = 2/2 + 0.01/0.01 = 2 (lack of synergy/antagonism).

The lack of synergy may be due to the fact that each remedy is effective individually. Therefore, it is not possible to improve the effect significantly. *E. coli* can be effectively suppressed with both antibiotics and nanoparticles.

#### 2.3.4. *Serratia marcescens*

The *S. marcescens* bacterium demonstrated relatively good resistance to silver nanoparticles—inhibition was observed only at a concentration of 50 μg/mL. At the same time, antibiotic inhibition was at an average level. Different concentrations of AgNPs (0.1 μg/mL, 0.5 μg/mL, 1 μg/mL, 2 μg/mL, 10 μg/mL and 25 μg/mL) and ciprofloxacin (0.025 μg/mL, 0.05 μg/mL, 0.1 μg/mL, 0.25 μg/mL, 0.5 μg/mL, 1 μg/mL, 2 μg/mL and 5 μg/mL). The synergistic effect was insignificant, as can be seen from the index.
FIC I = 5/25 + 0.05/0.05 = 1.25 (lack of synergy).

A visualization of the partial addition of effects can be seen in Figure 7. As a recommendation for use, it can be suggested to enhance the effect of an antibiotic with a low concentration of silver nanoparticles. Conversely, the addition of small doses of an antibiotic improves the effect of using a significant concentration of silver nanoparticles. The synergistic effect is observed with unequal application and disappears completely in the middle part of the sample of values.

#### 2.3.5. *Staphylococcus aureus*

Staphylococcus demonstrated the greatest resistance to nanoparticles, slightly exceeding that of *S. marcescens. S. aureus* was quite susceptible to penicillin antibiotics. Different concentrations of AgNPs (2 μg/mL, 5 μg/mL, 10 μg/mL and 25 μg/mL) and amoxicillin (0.01 μg/mL, 0.025 μg/mL, 0.05 μg/mL, 0.1 μg/mL, 0.25 μg/mL, 0.5 μg/mL, and 1 μg/mL). A weak synergistic effect of these substances was observed at almost all concentrations; an example can be seen in Figure 7.
FIC I = 10/25 + 0.1/0.25 = 0.8 (lack of synergy/weak effect).

Here, the combined use of nanoparticles and antibiotics is recommended.

#### 2.3.6. *Candida albicans*

Using disco diffusion analysis, the comparative resistance of *C. albicans* to tetramethylenediethylenetetramine was previously shown. These data were confirmed when constructing growth curves, and these yeasts also turned out to be the most resistant to silver nanoparticles. Different concentrations of AgNPs (2 μg/mL, 5 μg/mL, 10 μg/mL, 25 μg/mL, and 50 μg/mL) and tetramethylenediethylenetetramine were studied (0.1%, 0.5%, 1%, 2%, 5%, 10%, and 35%). At the same time, a synergistic effect was observed, which was confirmed by the index:FIC I = 5/25 + 5/10 = 0.7 (partial synergistic effect).

The addition of effects was well expressed on the growth curves (Figure 8).

#### 2.3.7. *Candida serelytica*

The least resistant yeast species turned out to be *C. lipolytica*, both to silver nanoparticles and tetramethylenediethylenetetramine. Different concentrations of AgNPs (1 μg/mL, 2 μg/mL, and 5 μg/mL) and tetramethylenediethylenetetramine were studied (0.05%, 0.1%, 0.25%, 0.5%, and 1%). It was found that low concentrations of the fungicide had little effect on yeast, which, apparently, is a distinctive feature of this species. As in the previous case, there was a slight synergistic effect:FIC I = 0.1/0.25 + 1/2 = 0.9.

In some cases, the synergy effects were very clear (Figure 8).

#### 2.3.8. *Saccharomyces cerevisiae*

Brewer’s yeast was characterized by moderate resistance to fungicide and nanoparticles, while a significant synergistic effect was observed, which is confirmed by the calculation of the index:FIC I = 0.05/0.5 + 5/25 = 0.3.

Different concentrations of AgNPs (2 μg/mL, 5 μg/mL, 10 μg/mL, 25 μg/mL, and 50 μg/mL) and tetramethylenediethylenetetramine were studied (0.01%, 0.025%, 0.05%, 0.1%, 0.5%, 1%, 2% and 5%). The uniformity of dose-dependent effects of substances was expressed in contrast to the sharply decreasing effect in *C. lipolytica*. The addition of inhibitory effects is demonstrated in Figure 8.

### 2.4. Atomic Force Microscopy Study Results

#### 2.4.1. Effect of Ciprofloxacin and Silver Nanoparticles on *S. marcescens* Biofilms

Using atomic force microscopy, it was possible to visually compare the closeness of *S. marcescens* biofilms. The general trend is illustrated in Figure 9: biogenic and chemically synthesized silver nanoparticles in a significant concentration (25 μg/mL) had practically no effect on biofilms. Preparations with an antibiotic, even in a small concentration (0.1 μg/mL), had a noticeable effect on biofilms, regardless of the presence of nanoparticles. This is probably due to the fact that nanoparticles work more efficiently in suspension than on the surface. Bacterial cells of the first layer, apparently dying at an average concentration, can make it possible to form a biofilm on their surface, after which they no longer have a contact area with bacteria. This is not observed when using a water-soluble antibiotic. There is also no such effect at very high concentrations of nanoparticles (about 500 μg/mL)—bacteria cannot grow on such a surface (Figure 9).

#### 2.4.2. Changes in the Morphology of *E. coli* and *S. marcescens* Cells When Exposed to AgNPs

In Figure 10 and Figure 11, visual differences between undisturbed bacterial cells and cells after interaction with silver nanoparticles can be noted: nanoparticles led to shrinkage of *E. coli* and *S. marcescens* cells. *Sphagnum* extract had no pronounced effect except for the adhesion effect observed in *S. marcescens*. The surface structures of *E. coli* and *S. marcescens* changed slightly.

## 3. Discussion

The study of the bactericidal and fungicidal effects of silver nanoparticles of various characteristics, as well as the mechanisms of their action, is the subject of numerous studies. The search is underway for their most suitable conditions and concentrations for use against pathogens.

There is evidence that ionic silver is more effective than nanoparticles, and the main toxic effect of AgNPs on the bacterium *Pseudomonas putida* was caused by a certain amount of Ag^+^ ions released into the culture medium [64]. Moreover, if the particle size is within 10–20 nm, the toxicity is due to the penetration of nanoparticles into the bacterial cell and interaction with intracellular contents. When the size of nanoparticles exceeds 20 nm, the effect is due to silver ions [65]. In our study, a mixture of different fractions was used, both large (50 nm) and smaller (10 nm) nanoparticles. Therefore, apparently, both of these effects took place. Adhesion and accumulation of AgNPs on the cell surface is especially widespread in gram-negative bacteria. Penetration into bacterial cells occurs through water-filled channels called porines in the outer membrane [66]. It is also beneficial that silver nanoparticles can prevent the formation of bacterial biofilms [67] and sometimes more effectively than standard antibiotics [68]. For ESKAPE bacteria, such biofilm inhibition can range from 20 to 70% [69]. Our experiments using a replicator stamp have shown that at high concentrations of nanoparticles (up to 1000 μg/mL), inhibition of biofilm growth can reach 100%.

There is evidence of the effect of silver nanoparticles on various fungi. When they were added to the culture of mold fungi, the dry weight of the mycelium changed, and silver ions accumulated in it. Shortening and densification of hyphae, numerous changes in organelles and nuclei, and even cell plasmolysis were noted [70]. In relation to yeast *Candida* spp. it was also noted that silver nanoparticles caused swelling and peeling of the cell wall and reduced the enzymatic activity of proteinases and phospholipases [71]. The fungicidal effect of nanoparticles also depended on their size, and small (7 nm in size) quasi-spherical particles turned out to be especially effective [72]. The use of silver nanoparticles can be considered not only as an alternative to antibiotics or fungicides. One of the possible methods could be a combination of them to achieve a synergistic effect. Such studies have already been conducted; for example, for ampicillin, gentamicin, kanamycin, streptomycin, and vancomycin, a synergistic effect with AgNPs has already been shown [73]. In addition to antibiotics, other reinforcing components can be used, such as bacterial metabolites, which can improve the fungicidal effect against yeast of the genus *Candida* [74]. Oxidative stress factors, combined with exposure to silver nanoparticles, also demonstrated a synergistic effect against *C. albicans* and other yeasts [75]. It is also promising to use a combination of nanoparticles of various metals, such as silver and copper. Copper ions contribute to the oxidation of non-oxidized silver, which makes it possible to obtain its more active ionic form, causing a stronger antibacterial effect. It has also been suggested that various effects are combined: silver affects the plasma membrane of bacteria, while copper denatures nucleic acids and other internal biomolecules and cellular structures [76].

As for the inhibitory concentrations obtained by us, these data are partially consistent with the literature sources. In studies by Gulbagca et al. [77], nonparvilarly spherical biogenic silver nanoparticles ranging in size from 10 to 50 nm were effective against *S. aureus* and *E. coli* bacteria at a concentration of 256 μ/mL and against *ly* at a concentration of 128 μ/mL. Das and Velusamy [78] obtained data on the inhibition of *C. albicans* growth on solid media at a concentration of 100 μ/mL. Vijayan et al. stated that chitosan-stabilized biogenic silver nanoparticles exhibit significant antifungal activity against *C. albicans* at a concentration of 50 μ/mL [79]. Spherical AgNPs with a size of 3–30 nm effectively inhibited *C. albicans* at a concentration of 1000 μ/mL [80]. There is also evidence of the synergistic activity of silver nanoparticles and antibiotics, as well as fungicides. AgNPs (1000 μ/mL) mixed with an antibiotic (bacitracin, 10 μ/mL; ciprofloxacin, 10 μ/mL; tetracycline, 30 μ/mL; and cefixime, 5 μ/mL) in a 1:1 ratio were effective against *S. aureus* and *E. coli*. When mixing nanoparticles (2000 μ/mL) with antifungal agents (fluconazole, 150 μ/mL; metronidazole, 125 μ/mL) in a 1:1 ratio, an effect was observed against *C. albicans* [81]. Different methods of synthesis and stabilization of nanoparticles are used in different studies. This makes it possible to obtain particles of different sizes and properties. The effective concentrations vary from one researcher to another, but it becomes clear that nanoparticles at a concentration of 100–1000 μ/mL are quite effective against bacteria and fungi. When combined with antibiotics and antifungal agents, lower concentrations will be effective, as shown in our study.

## 4. Materials and Methods

### 4.1. Biogenic Synthesis of Silver Nanoparticles Using Sphagnum Moss Extract

Plants of *Sphagnum fallax* (H. Klinggr.) H. Klinggr. were taken from the oligotrophic swamp of the Inzensky district of the Ulyanovsk region, Russia. The washed and dried plants were placed in a flask with distilled water, previously brought to a boil. The contents of the flask were boiled for 5 min, after which the extract was settled in sterile conditions of the laminar box. The additives selected after centrifugation (for 5 min at a frequency of 2000 rpm) were filtered (pore size 0.45 microns) and stored in the dark at a temperature of 4–5 °C. The second component for the synthesis was a sterile solution of silver nitrate—AgNO_3_ (1 mmol/L). The two components were poured into one container under sterile conditions at room temperature and sealed. The proportions in which the *sphagnum* extract and AgNO_3_ were mixed were 1:2 by volume, respectively. The process of completing the synthesis of silver nanoparticles was controlled by changing the color of the reaction solution from transparent to light brown (Figure 12). The optimal time was 7 days in the dark; however, with artificial lighting, the process was faster—in 2 to 3 days. The suspension was centrifuged three times, washed, and concentrated to 1000 μg/mL. Our group has previously synthesized biogenic silver nanoparticles using *sphagnum* extract [82]. In our work, a different technique was used without heating the reaction mixture to 90 °C.

Chemical silver particles for a comparative experiment were obtained by the citrate method [83]. An aqueous solution (90 mg, 500 mL) of AgNO_3_ was heated to 100 °C on a magnetic stirrer; 10 mL of 1% sodium citrate was added while stirring. The color of the solution changed 10 min after the addition of sodium citrate. Biogenic nanoparticles were used in almost all experiments.

### 4.2. Characteristics of the Obtained Nanoparticles

Ultraviolet-visible spectroscopy was performed to confirm the presence of silver in the initial colloid. The optical density of biogenically obtained silver nanoparticles and *sphagnum* extract in the wavelength range of 200–900 nm was measured on a spectrophotometer (Figure 13). Spectrophotometric studies have shown the presence of a characteristic peak at a wavelength of 420 nm, which confirms the formation of colloidal silver nanoparticles (Figure 13a); in addition, there are peaks in the region of 200–220 nm and 240 nm, as well as noise in the region of 800–900 nm.

Samples of the obtained biogenic silver nanoparticles were studied using transmission electron microscopy (Figure 14A). It can be noted that the nanoparticles were rounded, oval, or irregular in shape, of different fractions, and the sizes ranged from 10 to 50 nm. Using elemental analysis of the image, it was shown that a high Ag peak is present in the region of the rounded particle, which confirms the silver nature of the nanoparticles (Figure 14B).

By the method of dynamic light scattering (DLS) and Doppler laser velocimetry, the hydrodynamic diameter (Dh = 238.0 ± 8.45 nm) and zeta potential (ζ = −27.9 ± 2.52 mV) were determined, characterizing the size and charge of biogenic particles at a temperature of 25 °C and pH 4.6–5. The average value of the hydrodynamic diameter of synthesized chemical particles was 59.35 ± 0.41 nm; the surface charge is −33.1 ± 0.5 mV.

The results of the study using dark-field microscopy showed that biogenic particles are stable in an aqueous environment. Data on chemically obtained silver nanoparticles and their stability were published by our scientific group earlier [84].

### 4.3. The Research Objects

Strains of opportunistic bacteria from the ESKAPE list were taken as microbiological objects: *Staphylococcus aureus* 6583 and *Escherichia coli* OP-50, as well as an opportunistic species mentioned earlier as a frequent causative agent of hospital infections—*Serratia marcescens* 9986. The soil bacterium *Pseudomonas putida* (Trevisan 1889) Migula 1895 was taken as a convenient, close analog of *P. aeroginosa*, and the marine bacterium *Alcanivorax borkumensis* SK2 was taken as an alternative.

The work with yeast was performed on three species: the most common *Candida albicans*, an alternative candida, *Candida lipolytica*, and harmless brewer’s yeast *Saccharomyces cerevisiae* (Figure 15). It is possible to note the white (alpha) morphology of cells without chlamydospores.

### 4.4. Methods of Research on Antibacterial and Antifungal Effects

These strains of microorganisms were cultured according to a standard procedure using liquid nutrient media (Lysogeny Broth, Marine Broth, and Nutrient Broth) in an incubator at a constant temperature (37 °C, 27 °C, 28 °C, and 30 °C) and stirring. Bacterial suspensions were reduced to either an optical density of 0.8–1 conventional units or 0.5 according to the McFarland standard, which corresponded to a concentration of about 1.5 × 108 CFU/mL. Yeast was grown similarly, using Saburo liquid nutrient medium, in an incubator at a constant temperature (33 °C, 28 °C, and 32 °C) and stirring.

For experiments with bacteria, six antibiotics from various groups were used: streptomycin [1], tetracycline [2], ceftriaxone [3], ciprofloxacin [5], erythromycin [6] and amoxicillin [7]. Copper sulfate (M), chloramine B (X), and Desgrane (an industrial agent consisting of a 35% solution of tetramethylenediethylenetetramine, D) were taken as fungicides.

To determine the inhibitory concentration of biogenic AgNPs, as well as to identify the most effective antibiotics and fungicides, disco-diffusion analyses were performed for each type of bacteria and yeast in Petri dishes on solid agarized media (Muller-Hinton and Saburo). The effect was determined by the width of the inhibition zone.

Further studies were conducted only with the most effective antibiotic or fungicide for each type.

A comparative analysis of the antibacterial and fungicidal properties of biogenic silver nanoparticles (including when combined with an antibiotic and a fungicide) was carried out by constructing 48-h growth curves of planktonic cultures of bacteria and yeast. In 96-well plates, 20 µL of a suspension of silver nanoparticles, 20 µL of an antibiotic or fungicide solution, and 20 µL of a suspension of microorganisms were added to each well with a volume of 200 µL. The plates were cultured with continuous intensive shaking and optical density measurement at a wavelength of 595 nm every hour for 48 h. The construction of the growth curve and statistical data processing were carried out in the Microsoft Excel 2016 program. The minimum inhibitory concentration (MIC) was determined by comparing the growth curves with the control values.

To determine the synergistic effect of a combination of two substances, the fractional inhibitory concentration index (FIC I) was calculated according to the formula [85]:FIC I = (FIC AB/MIC AB) + (FIC AgNPs/MIC AgNPs),
where FIC AB and FIC AgNPs represent fractional inhibitory concentrations of antibiotics and silver nanoparticles when used in combination, and MIC AB and MIC AgNPs represent minimal inhibitory concentrations of antibiotics and silver nanoparticles. Based on the obtained values of FIC I, the effect of the combination of nanoparticles and an antibiotic was evaluated as follows: synergistic (FIC I ≤ 0.5), partial synergistic effect (0.5 < FIC I ≤ 0.75), no effect (0.75 < FIC I ≤ 1.5) and antagonistic (FIC I ≥ 2).

The lethal concentration of biogenic silver nanoparticles was determined by the replication method, which completely excludes the formation of biofilms on solid agarized media. Bacterial cultures were inoculated on the surface of Petri dishes with agarized media. Using a 96-pin replicator stamp, a replica was made from a 96-well tablet filled with AgNPs of various concentrations. For convenience, some of the pins were previously deleted. Thus, in one replica, it was possible to apply about 50 drops of silver nanoparticles of different concentrations (from 5 to 1000 μg/mL) to a Petri dish. In the presence of a prolonged bactericidal effect, this prevented bacterial biofilms from developing at the places of contact with the droplet.

In addition, a more detailed comparison of the effectiveness of biogenic silver nanoparticles and a mixture of antibiotics with silver nanoparticles obtained by chemical method on the formation of biofilms by the studied microorganisms was carried out, using the example of *S. marcescens*. The preparations were studied using atomic electron microscopy methods, and the change in the morphology of biofilms after cultivation was evaluated. The bacteria were cultured in liquid media in an incubator, without stirring, in 24-well plates with a 9 mm cover glass at the bottom. Silver nanoparticles at a concentration of 25 μg/mL and/or an antibiotic were added to them (0.1 μg/mL). After 48 h, the wells were washed three times with a phosphate buffer, fixed with 1% glutaraldehyde, and washed with distilled water. The glasses were removed by gluing them to a slide. Further, the preparations were studied using an atomic force microscope.

Atomic force microscopy was also used to determine the effect of *sphagnum* extract and silver nanoparticles on the morphology of planktonic forms of *E. coli* and *S. marcescens*. The bacteria were cultured in liquid media in an incubator, with stirring. Silver nanoparticles at a concentration of 10 μg/mL (and 5 μg/mL for *E. coli*) or *sphagnum* extract were added to them. After 24 h, the drops were fixed with 1% glutaraldehyde, washed with distilled water, dried on a slide, and studied using an atomic force microscope.

## 5. Conclusions

(1)The synthesis of biogenic silver nanoparticles using an aqueous extract of *sphagnum* moss (*Sphagum fallax*) was applied for the first time. Stable biogenic silver nanoparticles of rounded and irregular shape, ranging in size from 10 to 50 nm, with pronounced antimicrobial and antifungal effects, have been obtained.(2)The effectiveness of different concentrations of nanoparticles against opportunistic bacteria and yeast was analyzed (in two-day experiments). For *S. aureus* and *E. coli*, the effect appears at 50 μg/mL; growth suppression occurred at 100 μg/mL. For the bacterium *S. marcescens*, the effect appeared at 25 μg/mL, but growth suppression occurred in the range from 200 to 500 μg/mL. The antifungal effect of nanoparticles on *C. albicans* appeared at 100 μg/mL; complete suppression occurred at 300 μg/mL.(3)The possibilities of reducing the dosages of antibiotics and silver nanoparticles when used together against opportunistic bacteria were analyzed. When analyzing small concentrations, the greatest synergistic effect was observed at *S. aureus* and *P. putida*; an insignificant effect was found at *S. marcescens* and *A. borkumensis*, and there was no such effect at *E. coli*. The most resistant species to silver nanoparticles were *S. aureus* and *S. marcescens*, and the least—*E. coli* and *A. borkumensis*. *E. coli* turned out to be the species with the greatest vulnerability, both to nanoparticles and to an antibiotic.(4)In three species of yeast (*C. albicans*, *C. lipolytica,* and *S. cerevisiae*), a synergistic effect was observed with the combined use of silver nanoparticles and a fungicide; the most effect was at the brewer’s yeast. The most resistant species to silver nanoparticles turned out to be *C. albicans*, and the least resistant was *C. lipolytica*.(5)Silver nanoparticles were much more effective in the thickness of the suspension when stirred than at rest, concentrated in the bottom layer.

As a result, it can be concluded that silver nanoparticles obtained by the presented method can be used as an independent remedy against *E. coli* and *A. borkumensis*. It is possible to enhance the effect of an antibiotic against *P. putida* and *S. aureus* using nanoparticles. An increased concentration of nanoparticles should be applied to *S. marcescens*. The use of silver nanoparticles in conjunction with a fungicide against opportunistic candida seems to be quite effective. The obtained data can become the basis for the development of methods for the combined use of various agents against pathogens based on their synergistic effect.

## Figures and Tables

**Figure 1 ijms-25-12494-f001:**
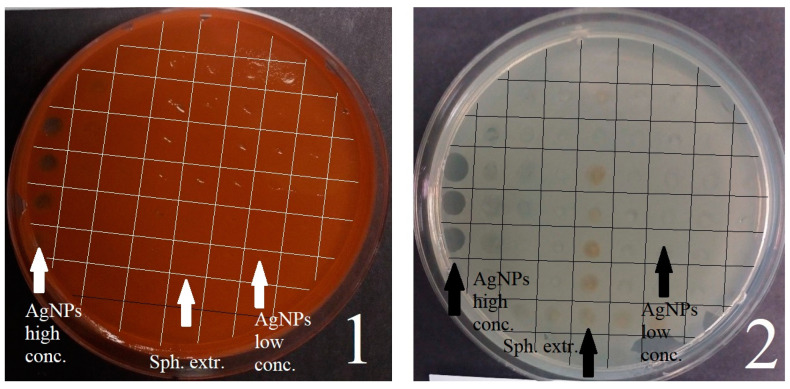
The results of replication of silver nanoparticles (of different concentrations) and *sphagnum* extract on the formation of biofilms of *S. marcescens* (1) and *P. putida* (2).

**Figure 2 ijms-25-12494-f002:**
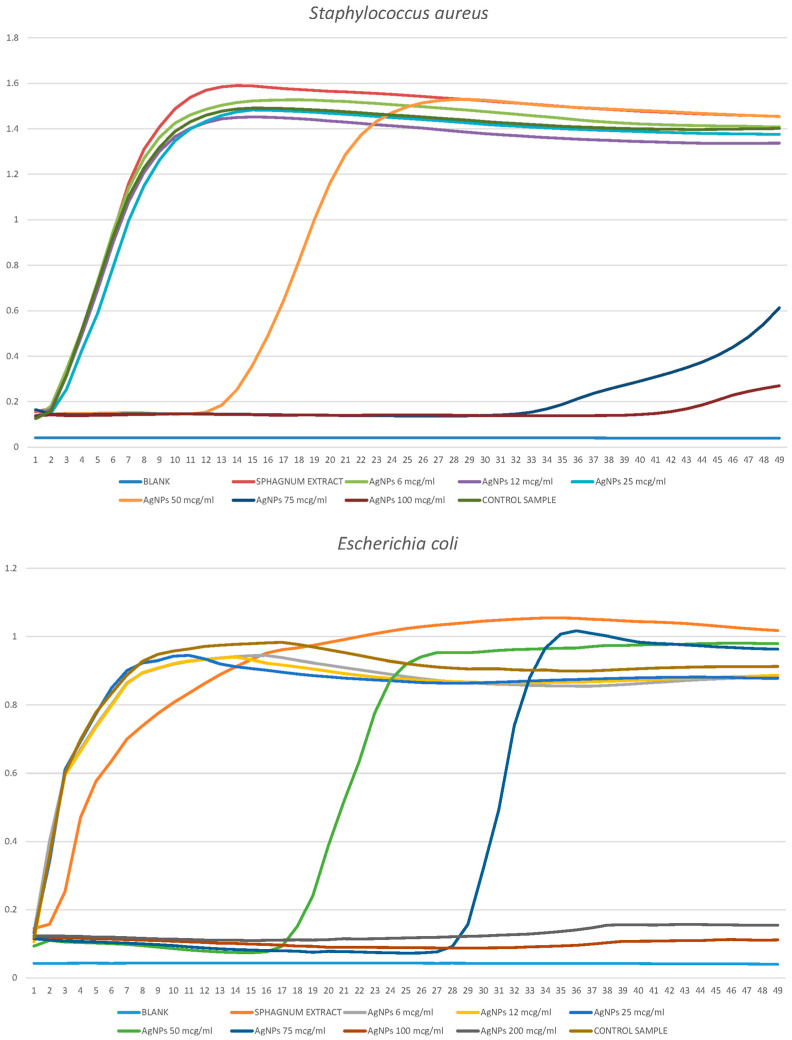
Determination of the antibacterial activity of biogenic silver nanoparticles and *sphagnum* extract against *S. aureus* and *E. coli* by the method of constructing growth curves. The concentrations of nanoparticles are given in the captions.

**Figure 3 ijms-25-12494-f003:**
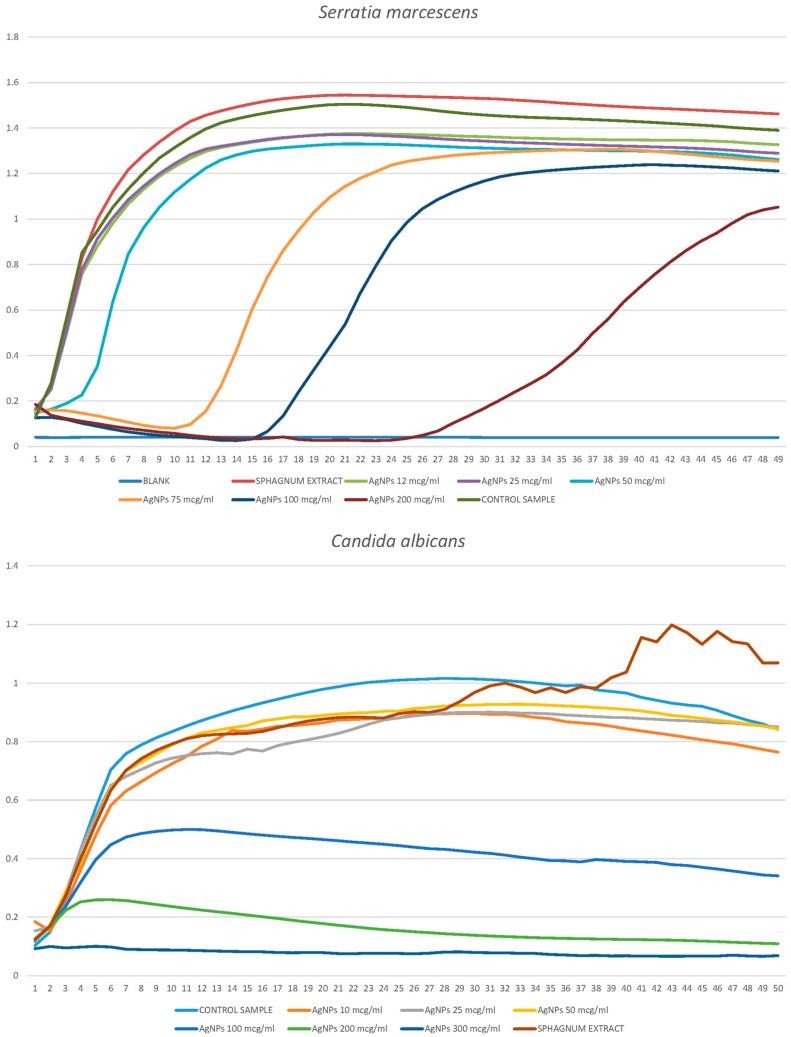
Determination of the antibacterial activity of biogenic silver nanoparticles and *sphagnum* extract against *S. marcescens* and *C. albicans* by constructing growth curves. The concentrations of nanoparticles are given in the captions.

**Figure 4 ijms-25-12494-f004:**
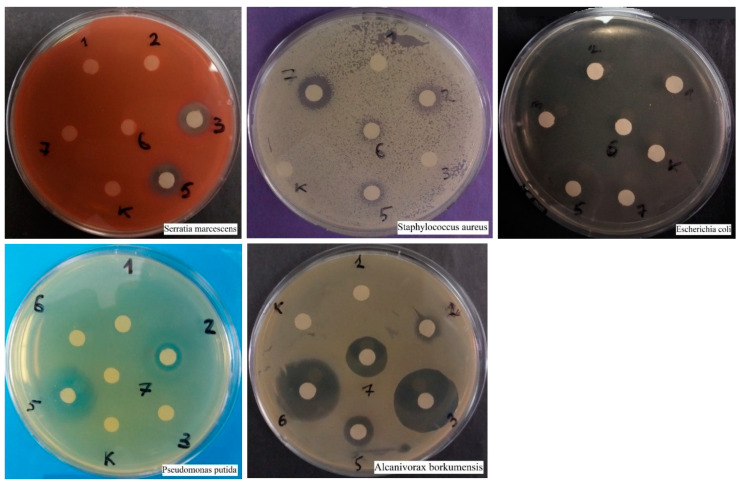
Visualization of the results of disco diffusion analysis of inhibition of 5 bacterial species by 6 antibiotics. 1—streptomycin, 2—tetracycline, 3—ceftriaxone, 5—ciprofloxacin, 6—erythromycin, 7—amoxicillin, K—control.

**Figure 5 ijms-25-12494-f005:**
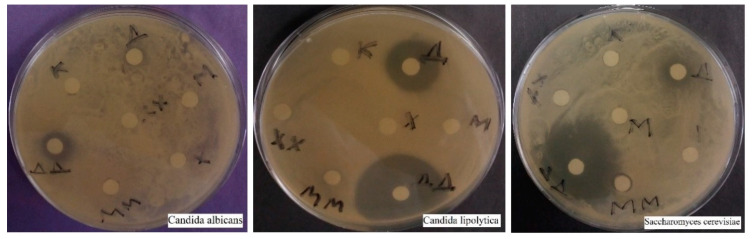
Visualization of the results of the disco-diffusion analysis of the effect of three fungicides on three types of yeast. M—copper sulfate, MM—10-fold copper sulfate, X—chloramine, XX—10-fold chloramine, Δ—desgrane, Δ Δ—10-fold desgrane, K—control.

**Figure 6 ijms-25-12494-f006:**
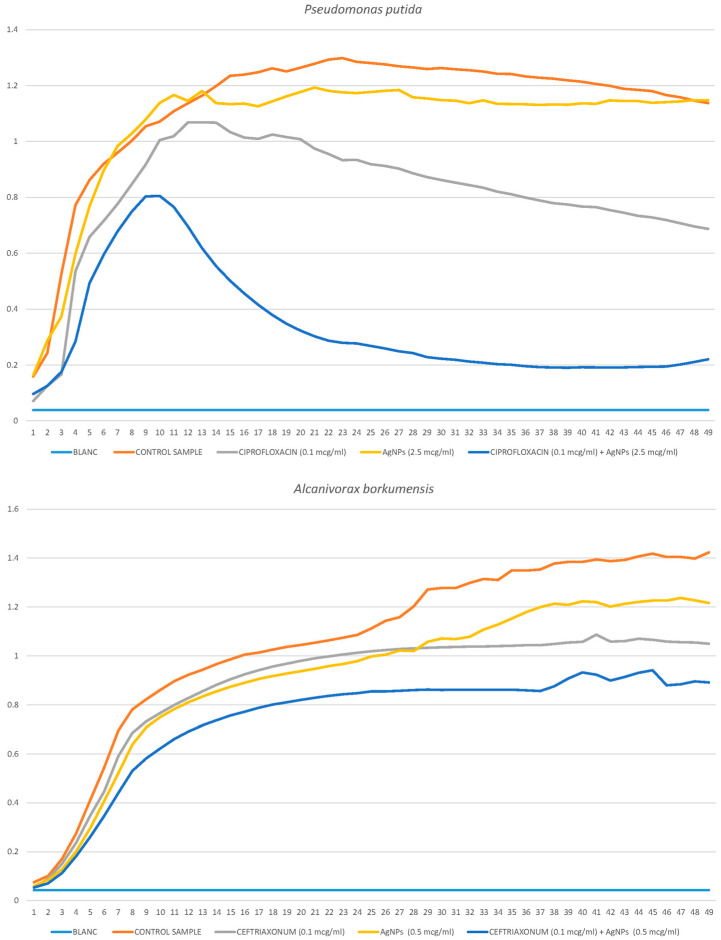
Growth curves of *P. putida* and *A. borkumensis* cultivated in the presence of the antibiotic (ciprofloxacin or ceftriaxone) and silver nanoparticles.

**Figure 7 ijms-25-12494-f007:**
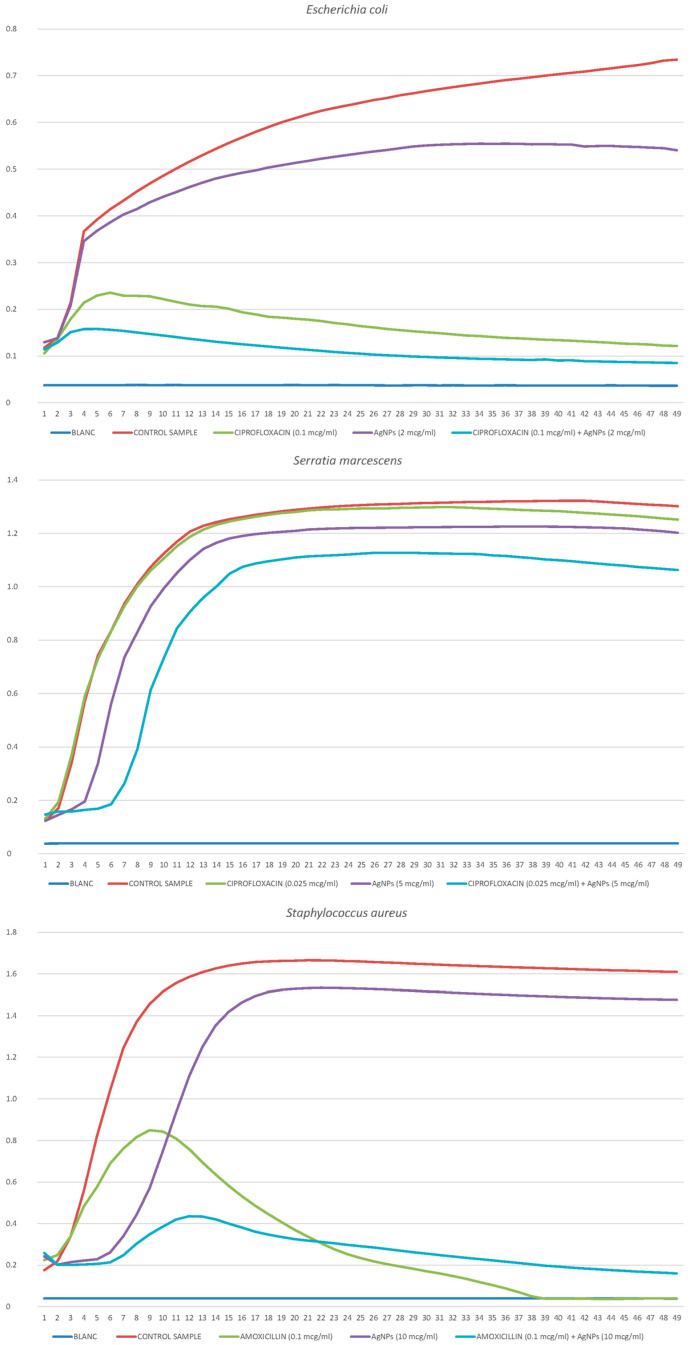
Growth curves of *E. coli*, *S. marcescens,* and *S. aureus* cultivated in the presence of the antibiotic (ciprofloxacin or amoxicillin) and silver nanoparticles.

**Figure 8 ijms-25-12494-f008:**
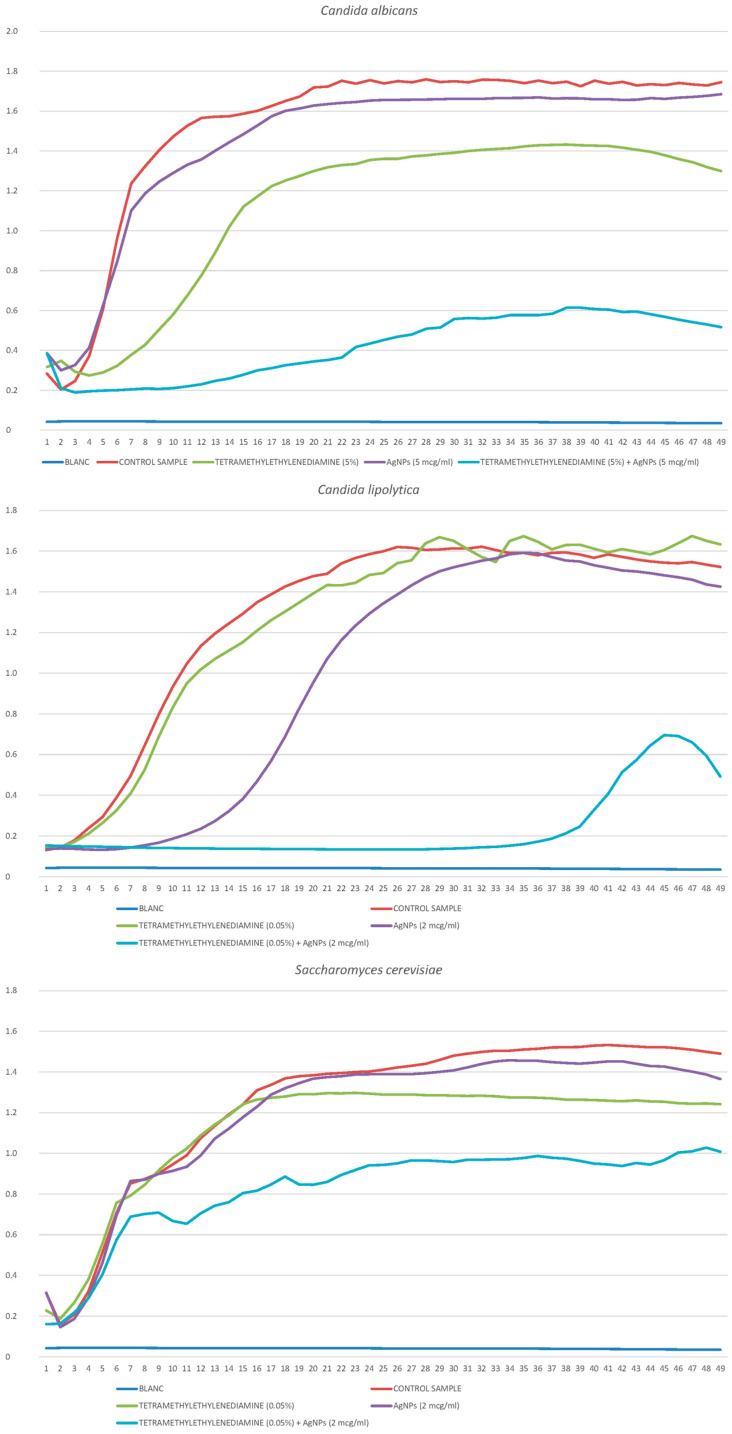
Growth curves of *C. albicans*, *C. lipolytica,* and *S. cerevisiae* cultivated in the presence of tetramethylenediethylenetetramine and silver nanoparticles.

**Figure 9 ijms-25-12494-f009:**
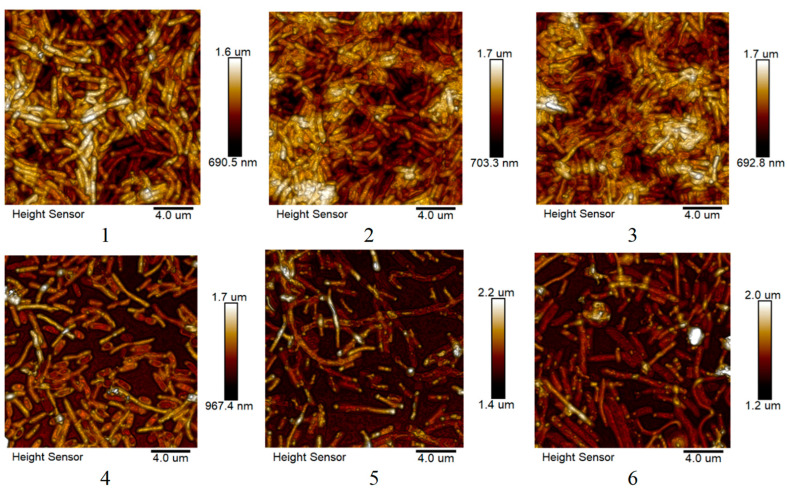
Images of *S. marcescens* biofilms obtained by atomic force microscopy cultured in the presence of silver and ciprofloxacin nanoparticles. 1—control sample, 2—biogenic silver nanoparticles, 3—chemically synthesized silver nanoparticles, 4—antibiotic, 5—biogenic silver nanoparticles with the addition of an antibiotic, and 6—chemically synthesized silver nanoparticles with the addition of an antibiotic.

**Figure 10 ijms-25-12494-f010:**
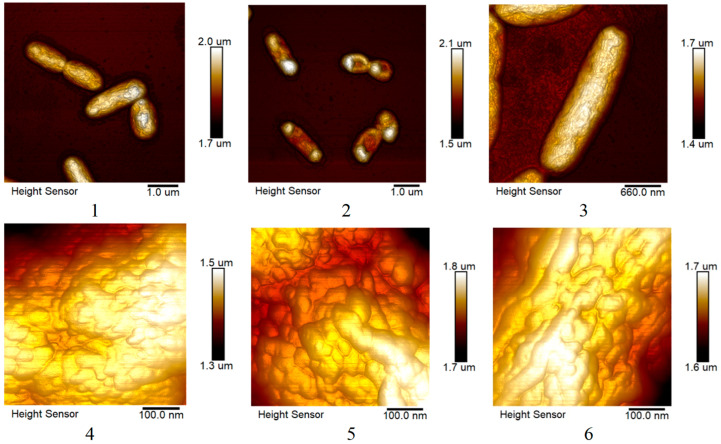
Images of the general appearance and surface of *E. coli* cells cultured in the presence of silver nanoparticles and *sphagnum* extract were obtained using atomic force microscopy. 1, 4—control sample, 2, 5—silver nanoparticles, 3, 6—*sphagnum* extract.

**Figure 11 ijms-25-12494-f011:**
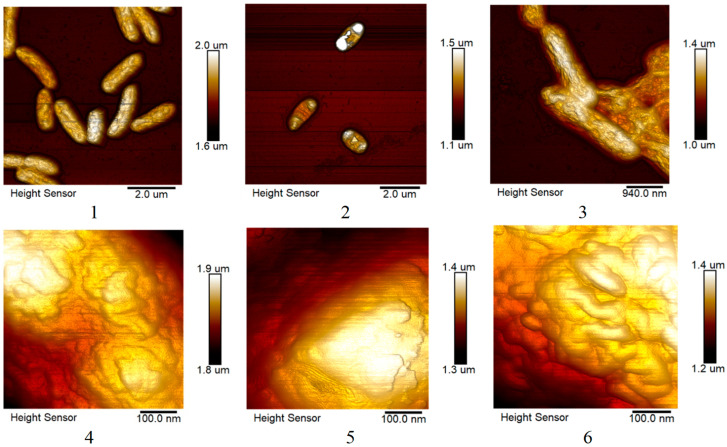
Images of the general appearance and surface of *S. marcescens* cells cultured in the presence of silver nanoparticles and *sphagnum* extract were obtained using atomic force microscopy. 1, 4—control sample, 2, 5—silver nanoparticles, 3, 6—*sphagnum* extract.

**Figure 12 ijms-25-12494-f012:**
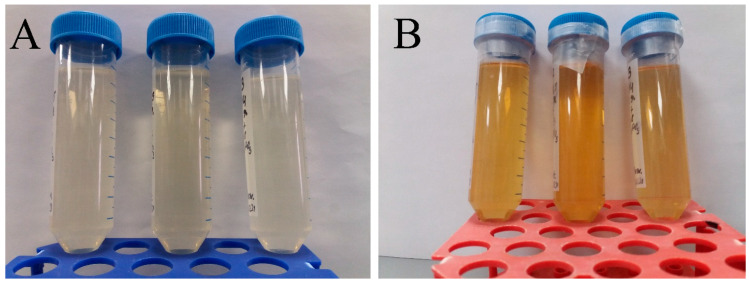
Changes in the color of the reaction mixture of moss extract and silver nitrate, from transparent at the beginning of synthesis (**A**) to yellow-brown after 7 days (**B**).

**Figure 13 ijms-25-12494-f013:**
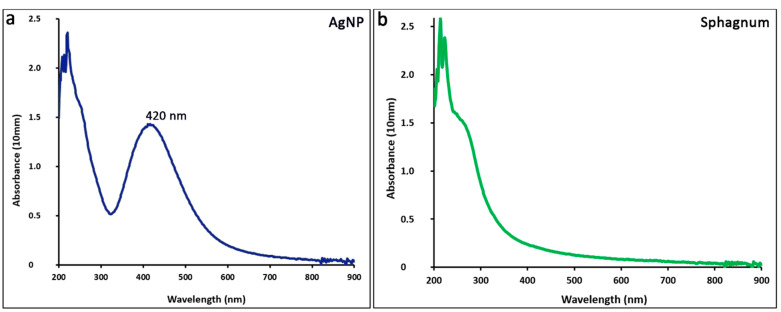
Absorption spectra in the UV-visible range of nanoparticle suspensions: biogenic silver nanoparticles (**a**); *sphagnum* extract (**b**).

**Figure 14 ijms-25-12494-f014:**
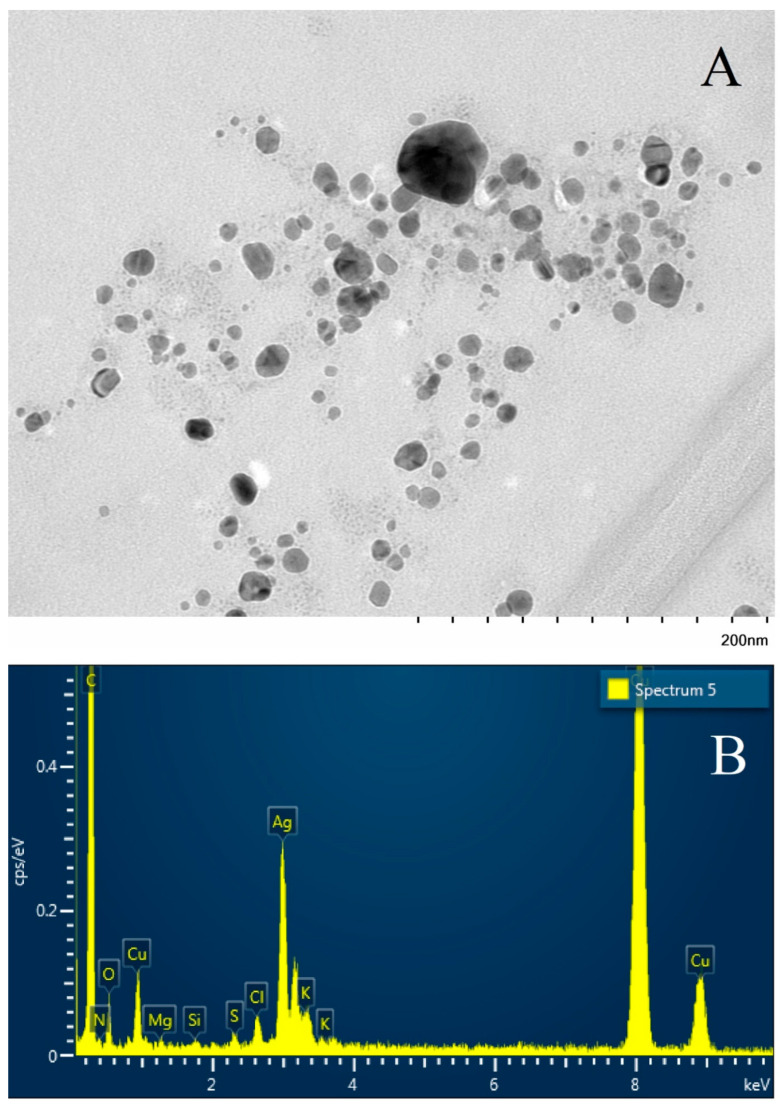
Visualization of AgNPs using transmission electron microscopy—(**A**). The shape of the particles is irregularly spherical. Elemental analysis of silver nanoparticles using transmission electron microscopy—(**B**).

**Figure 15 ijms-25-12494-f015:**
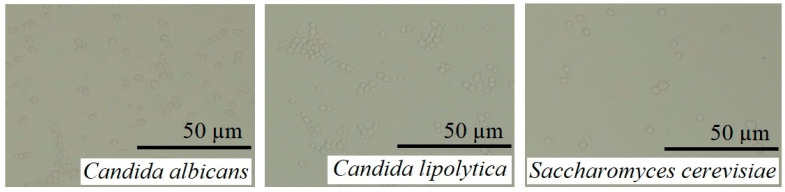
Microscopic photographs of planktonic yeast forms from fresh nocturnal cultures. Bright field microscopy.

**Table 1 ijms-25-12494-t001:** Results of the disco-diffusion analysis of the antibacterial and antifungal effect of silver nanoparticles of various concentrations, in comparison with *sphagnum* extract and control samples containing no substances. The numbers show the thickness of the inhibition zone in millimeters.

Active Substance	*E. coli*	*S. marcescens*	*S. aureus*	*C. albicans*
Control sample (water)	0	0	0	0
AgNPs (25 μg/mL)	0	0	0	0
AgNPs (50 μg/mL)	0.5	0	0	0
AgNPs (75 μg/mL)	1	1	1	0
AgNPs (100 μg/mL)	2	1	2	0
AgNPs (200 μg/mL)	3	2	3	1
AgNPs (500 μg/mL)	3	3	3.5	5
AgNPs (1000 μg/mL)	3	3	3.5	6
*Sphagnum* extract	0	0	0	0

## Data Availability

Data is contained within the article.

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
