# Peer review of "Antimicrobial and Antifungal Action of Biogenic Silver Nanoparticles in Combination with Antibiotics and Fungicides Against Opportunistic Bacteria and Yeast"

_ijms, 2024, doi:10.3390/ijms252312494_

Round 1

Reviewer 1 Report

Comments and Suggestions for Authors

1.    most of the abstract text is written like an introduction. Author should revise it and add brief information about the experiment methodology.

2.    Key words should be minimized.

3.       The organism’s scientific names should be italicized and give taxonomic information like family and order. Please follow it throughout the manuscript.

4.        The short forms like SARS and ESKAPE should be explained at first mentioning.

5.       The manuscript needs extensive English language editing.

6.       Conclusion should be minimized

Note: detailed comments are in the attached pdf file.

Comments on the Quality of English Language

Author Response

For reviewer 1 (corrections in green).

  1. most of the abstract text is written like an introduction.Author should revise it and add brief information about the experiment methodology.

- Added brief information about the methods.

  1. Key words should be minimized.

-Keywords are shortened. Only the types of bacteria that were not typical for these studies were left.

  1. The organism’s scientific names should be italicized and give taxonomic information like family and order.Please follow it throughout the manuscript.

-Species names are in italics.

  1. The short forms like SARS and ESKAPE should be explained at first mentioning.

-It was done.

  1. The manuscript needs extensive English language editing.

-Corrections have been made,thank you.

  1. Conclusion should be minimized

- The volume of the text of the conclusions has been partially reduced.

Reviewer 2 Report

Comments and Suggestions for Authors

In this manu, the synergistic effects arising from the silver nanoparticles-combined use with some antibiotics and fungicides against various bacteria and yeasts were studied. And effective concentrations of substances were established, recommendations for the studied pathogenic species were presented and the effect of destruction of the bacterial membrane was illustrated. However, the necessity and significance of this study are not so clear. And there is a lack of some important comparative experiments.

Major points

1.       The Introduction part is too long. It can be shortened. In addition, generally in the last paragraph of Introduction, there should be some brief descriptions about this study.

2.       In the Fig3A, the size range of biogenic AgNPs was wide. Please give the size distribution analysis based on TEM images. In addition, please give the TEM image of chemical AgNPs. The PDI value of DLS of two kinds of AgNPs should be provided.

3.       Based on TEM images of biogenic AgNPs, they are in irregular shapes and at a wide range of size distribution. So the quality of this synthesis method looked not so good. Some powerful explanations about choosing this method are required.

4.       In the Line 237-238, “The results of the study showed that biogenic particles were stable in the aquatic environment”. Please give some explanations about how it can get this conclusion based on DLS results.

5.       In all the experiments about antimicrobial and antifungal activity of AgNPs at least in Part 3.1, there should be a comparison between biogenic and chemical AgNPs. Otherwise, it would make no sense to carry out the biogenic synthesis of AgNPs.  

6.       In Part3.4, the important results of AgNPs-combined with antibiotics or fungicides should be provided since they can suggest some mechanical information.

7.       There are too many figures in the manu. Some similar data can be moved to Supplementary data.

Comments on the Quality of English Language

Moderate editing of English language is required.

Author Response

For reviewer 2 (corrections in brown).

  1. The Introduction part is too long.It can be shortened.In addition, generally in the last paragraph of Introduction, there should be some brief descriptions about this study.

-You're right. The introduction has been shortened. In accordance with the rules of the journal,Chapter4–Discussion has been added. This chapter now partially contains information from the introduction.Descriptions of ourwork have been added at the end of the introduction,at your direction.

  1. In the Fig 3A, the size range of biogenic AgNPs was wide.Please give the size distribution analysis based on TEM images.In addition, please give the TEM image of chemical AgNPs.The PDI value of DLS of two kinds of AgNPs should be provided.

-TEM images of chemical nanoparticles were removed from the previous version of the article to reduce its volume.

  1. Based on TEM images of biogenic AgNPs, they are in irregular shapes and at a wide range of size distribution.So the quality of this synthesis method looked not so good.Some powerful explanations about choosing this method are required.

- Biogenic synthesis of silver nanoparticles in most cases leads to the formation of particles of different calibers. This can be concluded from electronic images by different authors, for example,DOI:10.1016/J.heliyon.2019.e02980,DOI:10.5101/nbe.v5i1.p34-38,DOI:10.1007/s13204-015-0463-1,DOI:10.3389/fbioe.2021.652362.Wehaveaddedthesesources to the literature review. Explanations about the choice of method are given in the introduction. Simplicity and no need to use high temperatures and chemical reagents. Considering that a huge number of researchers obtain nanoparticles of different fractions using this method, we had no doubts. The data was well consistent with the work of our colleagues.

  1. In the Line 237-238, “The results of the study showed that biogenic particles were stable in the aquatic environment”.Please give some explanations about how it can get this conclusion based on DLS results.

- You are right, this conclusion was obtained based on other studies.We have reduced the volume of the article and have not included dark-field microscopy data.

  1. In all the experiments about antimicrobial and antifungal activity of AgNPs at least in Part 3.1, there should be a comparison between biogenic and chemical AgNPs.Otherwise, it would make no sense to carry out the biogenic synthesis of AgNPs.

- The mainpart of the work was carried out with biogenic silver nanoparticles (growth curves and a replicator stamp). Chemical nanoparticles were used only in the work with biofilms. As such, no comparison was made. The main task of this work was something else. We have studied, infact, only biogenic silver nanoparticles. This study makes sense, since biogenic silver nanoparticles are quite simple to synthesize. In addition, the comparison of chemical and biogenic nanoparticles should be comprehensive in order to exclude the influence of such a factor as particle size. For example, it may turn out that the difference in the bactericidal effect is due more to the particle size than to the synthesis method. Such a study includes a large number of options and this is a topic for a separate article. I repeat that the comparison of nanoparticles was not the main topic for us in this article.

  1. In Part 3.4, the important results of AgNPs-combined with antibiotics or fungicides should be provided since they can suggest some mechanical information.

-Unfortunately, we have not conducted a study of the combined effect of nanoparticles and antibiotics on biofilms. This is a topic for further research. Thank you for your interest.

  1. There are too many figures in the manu.Some similar data can be moved to Supplementary data.

-We agree with you. Most of the figures were removed from the original manuscript during the editing stage.We have left only the necessary ones.At the same time, some reviewers, on the contrary,asked to add some figures.

Reviewer 3 Report

Comments and Suggestions for Authors

Author Response

Sphagnum extract has been used to synthesized by this research team previously.The paper needs to be properly cited and it needs to stated in the introduction and methods if the particles are the same or if there is any difference.It needs to discussed if the size and morphology of the particles are similar to the previously published particles.o Akhatova, Farida, et al. "Comparative Characterization of Iron and Silver Nanoparticles: Extract-Stabilized and Classical Synthesis Methods." International Journal of Molecular Sciences 24.11 (2023): 9274.

-We have added a link to this article.

A figure displaying the DLS data should be included in the supplementary data and be referred to in the results section.

-We have given the data not in the form of a figure, but in the form of values of the zeta potential and the hydrodynamic radius (inChapter3.2.).

FIC is discussed extensively in Figures 10-12.o Citations for the thresholds for what is considered synergy (lines 289-294) lines . There are multiple papers where this can be cited.

-Thank you. A link to a similar work has been added.

During the AFM studies, was Root mean square (RMS) roughness measured?This measurement can be discussed as a sign of antimicrobial effect with the silver NPs as increased roughness correlates with decreased cell membrane integrity.

-No, such data has not been measured. In this article, AFM images were given to visualize the effects.

Author Contributions section needs to be updated.

- The section has been changed.

Reviewer 4 Report

Comments and Suggestions for Authors

This manuscript presented data on antimicrobial action by biogenetic silver nanoparticles. Use of silver nanoparticles for these purposes are widely known including combined use of other factors. Therefore, conceptual novelty of this work is not sufficiently high. Therefore, successful publication depends on the quality of the presented data. Unfortunately presented data have very low quality. The used silver nanoparticles look like dust. They are not well-sized controlled particles. In addition, these particles are not well characterized as seen in absence of data on particle size distribution and surface functionality etc. Biological data are just standard observation. Time course data re not well analyzed. In addition, comparisons of the obtained data over the other related data in the past literatures are poor. Therefore, superiority and advantages in this work are not so clear. With these negative factors, I do not recommend publication of this work in Int. J. Mol. Sci.

Author Response

Forreviewer 3 (corrections in pink).

This manuscript presented data on antimicrobial action by biogenetic silver nanoparticles.Use of silver nanoparticles for these purposes are widely known including combined use of other factors.Therefore, conceptual novelty of this work is not sufficiently high.

-Despite numerous studies, the introduction of the use of nanoparticles into medical practice is still far away. In our article, we tried to get closer to solving this problem by presenting quantitative data. We used different concentrations of active substances and proposed a new synthesis method. The bacterium A.borkumensis was used in such studies for the first time.

Therefore, successful publication depends on the quality of the presented data. Unfortunately presented data have very low quality. The used silver nanoparticles look like dust.

In order to excluded us from the study, in addition to imaging using electron microscopy, we conducted an elemental analysis. The high peaks of silver make it impossible to make a mistake –we are dealing with silver nanoparticles. In addition, the dust has a less saturated color in the pictures. You can see the results shown in Figure 14.

They are not well-sized controlled particles.In addition, these particles are not well characterized as seen in absence of data on particle size distribution and surface functionality etc.

-In Chapter 3.2. we tried to characterize nanoparticles by presenting visualization, size, shape, hydrodynamic radius and charge. These data give a general idea of which nanomaterial was obtained. The detailing would make sense if the exact separation of nanoparticles into fractions were carried out. Given that we propose a simple and practical method, one can hardly expect that such a separation will actually be carried out. Anyway, the result of our work was to test the effectiveness of nanoparticles obtained in an affordable way. The size heterogeneity that you mention will continue to occur in the future, when using this synthesis method.At the same time, the nanoparticles have shown effectiveness.

Biological data are just standard observation.

-Thank you for that remark. We have been striving for generally accepted standards in our research.

Time course data re not well analyzed.

-The presented growth curves (Figures 2, 3, 6, 7 and 8) show data from hourly measurements of the density of plankton colonies. These curves provide an understanding of how and at what point inhibition occurred. Its eems to us that this information is presented in sufficient detail.

In addition, comparisons of the acquired data over the other related data in the past literature sare poor.

-Thank you for your valuable comment.We have addedChapter3.5.with a comparison of our data with literary sources.

Reviewer 5 Report

Comments and Suggestions for Authors

Antimicrobial and antifungal action of biogenic silver nanoparticles in combination with antibiotics and fungicides, against opportunistic bacteria and yeast is very interesting paper. Some improvement is required.

Line 13: Nanoparticles (such as silver, gold, copper, ….) can be combined with antifungal agents in the same way.

Line 14: Biogenic silver nanoparticles synthesized using environmentally friendly technology using extracts of biologically active plants (which process?)

Line 106: Silver nanoparticles (AgNPs) could be such an instrument (what are particle size, specific surface area and shape)

Line 127: the process takes place at a low temperature (in which temperature range)

Line 169: The fungicidal effect of nanoparticles also depended on their size, and small (from 7 nm) particles. What is shape of these particles?

Line 228: It can be noted that the nanoparticles were rounded, oval or irregular in shape, of different fractions, and the sizes ranged from 10 to 50 nm. Are these particles agglomerated?

Line 240: Figure 3. Visualization of AgNPs using transmission electron microscopy – A. (Synthesis of silver nanoparticles was performed from silver nitrate. How to explain the presence of chlorine at Figure 3)

Line 370: What are units for the x-axis and Y-axis at the Figure 6

Line 490; In this case, (silver) nanoparticles are more effective to use without the addition of an antibiotic (why?)

Line 612: The effectiveness of different concentrations of nanoparticles against opportunistic bacteria and yeast was analyzed (in which time)

General questions:

What is a request for morphology and particle size of nanosilver in order to be applied in antimicrobial and antifungal action?

Line 608: The technique of synthesis of biogenic silver nanoparticles using an aqueous extract of sphagnum moss (Sphagum fallax) was applied for the first time. What are parameters for the synthesis of biogenic silver?

Author Response

For reviewer 4 (corrections in blue).

Antimicrobial and antifungal action of biogenic silver nanoparticles in combination with antibiotics and fungicides, against opportunistic bacteria and yeast is very interesting paper.Some improvement is required.

-Thank you for your appreciation of ourwork.

Line 13: Nanoparticles (such as silver, gold, copper, ….) can be combined with antifungal agents in the same way.

-We really did not specify which nanoparticles we are talking about. In this case, we assumed that it would be clear that we are talking about silver nanoparticles. Corrections have been made.

Line 14: Biogenic silver nanoparticles synthesized using environmentally friendly technology using extracts of biologically active plants (which process?)

- The process of biosynthesis of silver nanoparticles has been described in the literature many times, so we used its mention. The process itself is most often a mixing of precursors, one of which is a reducing agent (stabilizer) of plant origin. As are sult, silver changes from an ionic form to a molecular one. But the methods of such synthesis can be diverse.We have corrected the sentence according to your instructions, adding a mention of biosynthesis to it.

Line 106: Silver nanoparticles (AgNPs) could be such an instrument (what are particle size, specific surface area and shape)

-You are right, the characteristics of nanoparticles can play a crucial role. Next, we point to the sizes of the nanoparticles,which are indicated in the literature. A detailed description of the nanoparticles obtained by us is given in the Materials and Methods section.

Line127: the process takes place at a low temperature (in which temperature range)

- Thank you. Added a clarification. These processes are often carried out at room temperature.

Line 169: The fungicidal effect of nanoparticles also depended on their size, and small (from 7 nm) particles.What is the shape of these particles?

-We added information that the particles were quasi-spherical.

Line 228: It can be noted that the nanoparticles were rounded, oval or irregular in shape, of different fractions, and the sizes ranged from 10 to 50 nm.Are these particles agglomerated?

-Based on the electron microscopy data (Figure 14), the nanoparticles did not form agglomerates. However, it should be noted that with long-term storage (for several months) such clusters can form.

Line 240: Figure 3. Visualization of AgNPs using transmission electron microscopy – A. (Synthesis of silver nanoparticles was performed from silver nitrate.HowtoexplainthepresenceofchlorineatFigure3)

- Most likely, chlorine in a small amount got intot he NaCl salt, which wasin the buffer.

Line 370: What are units for the x-axis and Y-axis at the Figure 6

-You're right,it's worth signing.The X-axis – hours of the experiment and Y-axis – optical density of the suspension.

Line 490; In this case,(silver) nanoparticles are  more effective to use without the addition of an antibiotic (why?)

- The FIC value here is 1.4. This value corresponds to the almost complete absence of a synergistic effect. This suggests that instead of adding an antibiotic, you can achieve a greater effect by increasing the concentration of nanoparticles in the same proportions.

Line 612: The effectiveness of different concentrations of nanoparticles against opportunistic bacteria and yeast was analyzed (in which time)

-Thank you. We have added the time indicators of the experiment.

What is a request for morphology and particle size of nanosilver in order to be applied in antimicrobial and antifungal action?

- Taking into account our experience and the analysis of many literary sources, we can, with caution, state that the size of 5-10 nm is favorable, while nanoparticles of 30-35 nm are also effective. We described this issue in detail earlier in our review https://doi.org/10.3390/mi12121480.

Line 608: The technique of synthesis of biogenic silver nanoparticles using an aqueous extract of sphagnum moss (Sphagum fallax) was applied for the first time. What are the parameters for the synthesis of biogenic silver? 

- The parameters you are interested in are listed in Chapter 3.1. Thank you for your help and constructive questions to help improve our article.

Round 2

Reviewer 2 Report

Comments and Suggestions for Authors

1.       The TEM image of chemical AgNPs has not been provided yet. If the article volume is limited, it can move to supplementary data. In addition, the PDI values of DLS of two kinds of AgNPs

2.       The point about biogenic particles were stable in the aquatic environment was not well explained. If the article volume is limited, it can move to supplementary data.

3.       For any nanomaterial which is set to be applied, the quality control is vital. A new good synthesis method should not be only easy to carry out. The quality and performance materials from this synthesis method are the most important. It there is no comparison from chemical and biogenic AgNPs, we cannot conclude that the AgNPs from biogenic are definitely better than the ones from chemical. So the explanations about a comparison between biogenic and chemical AgNPs cannot be accepted.

Comments on the Quality of English Language

 Moderate editing of English language is required.

Author Response

Comment 1: The TEM image of chemical AgNPs has not been provided yes. If the article volume is limited, it can move to supplementary data. In addition, the PDI values of DOLLS of two kinds of AgNPs

Comment 2: The point about biogenic particles where table in the aquatic environment was not well explained. If the article volume is limited, it can move to supplementary data.

Responses 1-2: In our recently published article 
 (https://doi.org/10.3390/biom14060611 ) the information you need has been collected. Data on the stability of silver nanoparticles are presented in sub-chapter 2.4.3. Figure 2 shows microscopic images of the biogenic and chemically obtained silver nanoparticles synthesized by us. 
This is a new article from our research group, it has not been published yet at the time of submission. It contains data that we cannot repeat in our article. We did not take separate pictures of chemically synthesized silver nanoparticles in our work, since their comparison with biogenic ones was not the task of the study. This comparison might be interesting to you and the readers, so I added a link to the second article.

Comment 3: For any nanomaterial which is set to be applied, the quality control is vital. A new good synthesis method should not be only easy to carry out. The quality and performance materials from this synthesis method are the most important. It there is no comparison from chemical and biogenic AgNPs, we cannot conclude that the AgNPs from biogenic are definitely better than the ones from chemical. So the explanations about a comparison between biogenic and chemical AgNPs cannot be accepted.

Response 3: You are right, we cannot conclude that nanoparticles from biogenic products are definitely better than from chemical ones. This conclusion is not in our article. This is a topic of a separate discussion that was not raised in our article. We only used chemical nanoparticles in one experiment, all other experiments were conducted with biogenic nanoparticles. There is no full-fledged comparison here. This comparison would require additional variants of all experiments. We have done experiments with biogenic silver nanoparticles and the article talks about how effective they can be. Thank you for your interest and your help.

Reviewer 3 Report

Comments and Suggestions for Authors

1) Please review the manuscript for spelling and grammatical errors.

2) Figure 15 Bright field Microscopy, please increase the image resolution. Cells are hard to see.

Author Response

Comment 1: Please review the manuscript for spelling and grammatical errors.

Response 1: That's a valuable point, thank you. We have worked through the text once again, trying to remove all the mistakes made.

Comment 2: Figure 15 Bright field Microscopy, please increase the image resolution. Cells are hard to see.

Response 2: Thank you! You're right, they're hard to see. I changed the figure 15, now the yeast cells are larger.

Reviewer 4 Report

Comments and Suggestions for Authors

It is OK.

Author Response

Comment: It is OK.

Response: Thank you for your help in improving our article!

Round 3

Reviewer 2 Report

Comments and Suggestions for Authors

The TEM image in Fig14A showed the size distribution range was so wide. It's impossible that the size range was from 10 nm to 50 nm written in the manu. So please provide a better TEM image and give the statistic data of size distribution.

In addition, some figures are not so clear, Fig2,Fig3, Fig6, Fig7, Fig8, etc. Please provide new figures with high resolution.

Comments on the Quality of English Language

English is fine.

Author Response

Comment 1: 
The TEM image in Fig 14A showed the size distribution range was so wide. It's impossible that the size range was from 10 nm to 50 nm written in the manu. So please provide a better TEM image and give the statistic data of size distribution.

Response 1: 
In fact, we could provide
a higher-quality TEM image of nanoparticles, but it would still be an image with a larger size range. We did not manually select areas of the image with particles of the same size. In our opinion, this does not accurately reflect the real situation. The size range from 10 to 50 nm is mentioned in studies by other authors, for example, https://doi.org/10.3390/molecules27113525 or https://doi.org/10.1016/j.matchemphys.2023.127413 . Our data correlate with literary sources.

Comment 2: 
In addition, some figures are not so clear, Fig 2,Fig 3, Fig 6, Fig7, Fig8, etc. Please provide new figures with high resolution.

Response 2:
Thank you for your valuable comment. We have tried to improve the quality of the images. When converting from a PDF file, we used the TIF file type this time. The result is slightly more contrasting images with a higher resolution. Thus, we have replaced Figures 2, 3, 6, 7 and 8.
